A historical legacy of antibiotic utilization on bacterial seed banks in sediments

Madueño Laura 1
Paul Christophe 1
Junier Thomas 2
Bayrychenko Zhanna 1
Filippidou Sevasti 1
Beck Karin 3
Greub Gilbert 4
Bürgmann Helmut 3
Junier Pilar pilar.junier@unine.ch 1
1 Laboratory of Microbiology, Institute of Biology, University of Neuchatel , Neuchâtel , NE , Switzerland
2 Vital-IT group, Swiss Institute of Bioinformatics , Lausanne , Switzerland
3 Eawag, Swiss Federal Institute of Aquatic Science and Technology , Kastanienbaum , Switzerland
4 Institute of Microbiology, University Hospital and University of Lausanne , Lausanne , Switzerland
Kormas Konstantinos
Electronic publication date: 2018 Jan 3
Publication date: 2018
Volume: 6
Electronic Location ID: e4197
Received 2017 Oct 19; Accepted 2017 Dec 5
Copyright: ©2018 Madueño et al.
Copyright year: 2018
Copyright holder: Madueño et al.
License: This is an open access article distributed under the terms of the Creative Commons Attribution License, which permits unrestricted use, distribution, reproduction and adaptation in any medium and for any purpose provided that it is properly attributed. For attribution, the original author(s), title, publication source (PeerJ) and either DOI or URL of the article must be cited.
License URL: https://creativecommons.org/licenses/by/4.0/

Keywords: Antibiotic resistance, Endospores, Clostridia, Tetracycline, Sulfonamide, Sediments, Seed bank

Funding: The Swiss National Science Foundation Interdisciplinary grant CR23I2_162810 NFP72 grant 407240_167116 The Swiss National Science Foundation (Interdisciplinary grant CR23I2_162810) provided funding for Sevasti Filippidou and Thomas Junier. Helmut Bürgmann acknowledges support by the Swiss National Science Foundation (NFP72 grant 407240_167116). The funders had no role in study design, data collection and analysis, decision to publish, or preparation of the manuscript.

==============================
The introduction of antibiotics for both medical and non-medical purposes has had a positive effect on human welfare and agricultural output in the past century. However, there is also an important ecological legacy regarding the use of antibiotics and the consequences of increased levels of these compounds in the environment as a consequence of their use and disposal. This legacy was investigated by quantifying two antibiotic resistance genes (ARG) conferring resistance to tetracycline (tet(W)) and sulfonamide (sul1) in bacterial seed bank DNA in sediments. The industrial introduction of antibiotics caused an abrupt increase in the total abundance of tet(W) and a steady increase in sul1. The abrupt change in tet(W) corresponded to an increase in relative abundance from ca. 1960 that peaked around 1976. This pattern of accumulation was highly correlated with the abundance of specific members of the seed bank community belonging to the phylum Firmicutes. In contrast, the relative abundance of sul1 increased after 1976. This correlated with a taxonomically broad spectrum of bacteria, reflecting sul1 dissemination through horizontal gene transfer. The accumulation patterns of both ARGs correspond broadly to the temporal scale of medical antibiotic use. Our results show that the bacterial seed bank can be used to look back at the historical usage of antibiotics and resistance prevalence.

Introduction

The use of antibiotics to treat infectious diseases represents one of the major scientific achievements of the 20th century. Millions of lives have been saved since the introduction of antibiotics into general medical practice for the treatment of a large range of bacterial infections, as well as other medical procedures (Marti, Variatza & Balcazar, 2014). After the initial use of antibiotics in medicine, the utilization of antibiotics to increase agricultural productivity has become a common practice (Carlet et al., 2011). Although the positive effect of the so-called antibiotic era on human welfare is not disputed, increased awareness of the risks posed by poor antibiotic stewardship counterbalances this success. Nowadays it is becoming clear that the disposal of antibiotics in natural ecosystems can have far-reaching consequences (Baquero, Martinez & Canton, 2008). Recent studies on antibiotics and the emergence of resistance suggest that the function of antibiotics in nature cannot be explained solely within the paradigm of chemical weapons in which these compounds have been used since their industrialized production (Aminov, 2009; Aminov, 2010). Instead, antibiotics and determinants of resistance have been proposed to be a fundamental component of the ecology and evolution of microbial ecosystems. Most of the antibiotics used today are chemical derivatives of small bioactive molecules that might perform a multitude of functions (Taylor, Verner-Jeffreys & Baker-Austin, 2011). In nature these molecules are thought to be produced at very low concentrations (Martinez, 2008), and for example, a study conducted at sub-inhibitory concentrations with erythromycin and rifampicin has shown that this low concentrations of antibiotics can modulate not only growth but also bacterial metabolism (Goh et al., 2002). Therefore, antibiotics can be expected to modulate microbial interactions and regulate the dynamics of microbial communities (Martinez, 2008).

Although antibiotic resistance could potentially emerge anywhere and at any given time, the emergence of a resistance factor has been generally associated with some fitness cost, and therefore novel resistance genes are expected to be under strong negative selection pressure (Bengtsson-Palme, Kristiansson & Larsson, 2017). In this context, the industrialized production, use, and disposal of antibiotics is a relatively recent phenomenon that has presumably exerted a positive selective pressure for pathogens to develop antibiotic resistance either as a consequence of mutation or by horizontally acquiring naturally occurring antibiotic resistance systems (Blair et al., 2015; Taylor, Verner-Jeffreys & Baker-Austin, 2011). The increasing levels of antibiotic resistance in bacteria isolated from clinical samples is a problem that threatens health care systems worldwide (Wright, 2010). Therefore, understanding the effect of antibiotic use on the natural reservoirs of ARGs and analyzing this recent historical event (the antibiotic era) in terms of the levels of circulating antibiotic resistance genes (ARGs) are essential to develop a management strategy to reduce current and future risks.

ARGs were clearly present in microbial communities before the antibiotic era as shown by phylogenetic analysis of genes conferring resistance to different classes of antibiotics (Aminov & Mackie, 2007). Evidence from work conducted on ancient DNA in permafrost (D’Costa et al., 2011) and an isolated cave (Bhullar et al., 2012) also support the existence of resistance without human intervention. Given the presumed role of human activity in the levels of resistance in the environment, one can thus expect an increasing abundance of such genes in the past century. However, direct evidence for this is currently restricted to a limited number of studies. For example, soil archives from two regions in Europe clearly demonstrate a link between the history of antibiotic use and the increase in the abundance of various genes conferring resistance to a large range of antibiotics (Graham et al., 2016; Knapp et al., 2010). Furthermore, the analysis of soil records also demonstrated the interconnection between the medical and non-medical use of antibiotics, as well as the effect of changes in policy towards a more strict stewardship in the reduction of ARGs from natural pools (Graham et al., 2016).

Besides soils, aquatic ecosystems have been identified as a key ecological component driving the emergence, spread, and persistence of antibiotic resistance (Baquero, Martinez & Canton, 2008; Taylor, Verner-Jeffreys & Baker-Austin, 2011). Water constitutes a circulating path of antibiotic-resistant organisms from human and animal populations to the environment and back into these populations, via the connection between wastewater treatment and drinking water production, respectively (Baquero, Martinez & Canton, 2008). Lake sediments are a major concern because they are a main environmental end-point not only for bacteria, but also for ARGs and antimicrobial agents (Kümmerer, 2009). The high numbers of cells in sediments make resuspended sediment material a potential source of resistance determinants. At the same time, lake sediments are natural environmental archives. Thus, the study of the sedimentary record might provide insights into the historical legacy of the antibiotic era and the accumulation of ARG in the environment. Attempts to use DNA extracted from sediments to investigate antibiotic resistance in aquatic systems have been made (Thevenon et al., 2012), but suffer from uncertainty regarding the preservation of the environmental signal in the sediments. Sediment microbial communities are strongly shaped by the redox gradients experienced during early diagenesis, and it is therefore unclear how much of the originally resistant community, or of their resistance determinants, is preserved in deeper sediment layers, and how this relationship is affected by environmental factors. The use of microbial seed banks preserved in the sedimentary record as a proxy offers a likely solution to these problems.

The seed bank can be broadly defined as a reservoir of dormant cells that can potentially be resuscitated under favorable environmental conditions (Lennon & Jones, 2011). One of the defining features of dormant cells is their reduced metabolic activity (Driks, 2002), decreasing the uncertainty generated by environmental changes during sediment diagenesis (Vuillemin et al., 2016). In addition, dormant cells are more resistant to degradation than their actively growing counterparts (Abecasis et al., 2013). We have used the latter property to develop a specific extraction method to enrich DNA from spores as an example of dormant cell forms (Wunderlin et al., 2016; Wunderlin et al., 2014b). With this approach we have previously shown that one particular group of bacteria capable of dormancy (endospore-forming Firmicutes) can be used as paleoecological biomarkers of the impact of lake eutrophication on microbial communities in sediments (Wunderlin et al., 2014a). Using the same selective method we investigated if the historical antibiotic usage has affected the levels of ARG found in the natural seed bank bacterial community. The hypothesis in this case is that information regarding the abundance and frequency of ARGs as the consequence of antibiotic use will be reflected in the dormant cells deposited in the sediment, regardless of the presence of the antibiotics themselves or intrinsic selection by the environment. To test this hypothesis, we investigated the levels of two ARGs conferring resistance to two antibiotics that were introduced earlier in the antibiotic era and with diverging histories of use. The gene tet(W) is one of the genes conferring resistance to tetracycline, a class of broad-spectrum antibiotics isolated from Streptomyces spp. between 1947 and 1950, constituting one of the earliest classes of antibiotics described and used (Roberts & Schwarz, 2016). The second ARG studied here, sul1, is one of the genes conferring resistance to sulfonamide drugs, which were also among the earliest antibiotics discovered. However, in contrast to tetracycline, sulfonamide and its derivatives were obtained by systematic screening of chemically synthesized compounds (Aminov, 2010; Davies & Davies, 2010). The diverging histories of production and use of these two antibiotics, as well as, the differences in the mechanisms generating resistance, will allow to proof the concept of using the seed bank to investigate the legacy of human antibiotics history, as well as to develop a method to investigate the natural history of antibiotics in the environment.

Material and Methods

Site description and sampling

A sediment core was retrieved with a gravity corer (UWITEC, Mondstein, Au) in August 2011 in an inactive canyon (C1) on the eastern side of the Rhone delta in Lake Geneva (Switzerland) (CAN01, coordinates 559901-139859, 79 m depth, 105 cm). This core has previously been dated by creating an age model based on 137Cs (corresponding to the 1963–1964 atmospheric nuclear tests maximum fallout and the 1986 Chernobyl accident) and magnetic susceptibility, which allowed assigning years to the sediment depth (Wunderlin et al., 2014a). Additional environmental data was obtained from a second sediment core (CAN02, 559405-140504, 96 m depth, 107 cm) retrieved in parallel to the sediment used for biological analysis. This second core was split in two lengthwise halves for a sedimentological description and chemical analysis. Manganese and iron measurements were performed at the University of Barcelona by X-ray fluorescence using an AVAATECH XRF core scanner (2000 A, 10 kV and 30 kV) every 2 mm. Correlation between the two sediment cores was carried out by visual description, sediment color and texture and by comparing magnetic susceptibility (MS) and density core profiles in order to assign the manganese and iron profiles to the ages investigated with CAN01 (Wunderlin et al., 2014a).

DNA extraction

Total community DNA and DNA from the seed bank were obtained using an indirect extraction method. The extraction of cells from sediments was performed as previously described (Wunderlin et al., 2013). The cells extracted from 3 g of wet sediment were filtered onto two different 0.2 µm pore-size nitrocellulose filters (Merck Millipore, Darmstadt, Germany). In one of the filters (1.5 grams of sediment) a treatment to separate seed bank from vegetative cells was performed on the biomass collected on nitrocellulose filters, as previously described (Wunderlin et al., 2016; Wunderlin et al., 2014b). The first step consisted of the lysis of vegetative cells by heat, enzymatic agents (lysozyme) and chemicals (Tris-EDTA, NaOH, SDS). Further DNase digestion was used to destroy any traces of free DNA. DNA was then extracted from the pre-treated (seed bank DNA) and the second non-pre-treated filter (total community DNA) using a modified protocol with the FastDNA® SPIN kit for soil (MP Biomedicals, Santa Ana, CA, USA) (Wunderlin et al., 2013), in which the lysing matrix was submitted to two successive bead-beating steps. Supernatants from each bead-beating step were treated separately downstream according to manufacturer’s instructions. The two DNA extracts per filter were pooled by precipitation with 0.3 M Na-acetate and ethanol (99%), stored at −20 °C overnight and centrifuged for 1 h at 21. 460× g and 4 °C. Supernatant was removed and the pellet was washed with 1 volume of 70% ethanol and centrifuged for 30 min at 21.460× g and 4 °C. Supernatant was removed and the residual ethanol was allowed to evaporate at room temperature. DNA was re-suspended in 50 µl of PCR-grade water. DNA was quantified using Qubit® dsDNA HS Assay Kit on a Qubit® 2.0 Fluorometer (Invitrogen, Carlsbad, CA, USA). DNA yield varied from 1.6 to 16 µg DNA/g for the total community DNA, and 6–23 ng DNA/g sediment for the seed bank DNA.

Quantitative PCR on tet(W) and sul1 genes

Quantitative Taqman®-PCR on sul1 and tet(W) genes was performed in 384-well plates using a LightCycler®480 Instrument II (Roche, Basel, Switzerland). For sul1, the primers used were qSUL653f and qSUL719r with tpSUL1 probe (Heuer & Smalla, 2007). The reaction mix for sul1 consisted of 2 µL of DNA template (between 0.08 and 1.39 ng/µL for seed bank DNA and 10 ng/µL for total community DNA), 0.025 µM of each primer, 0.25 µM of TaqMan probe and 1 × TaqMan® Fast Universal PCR Master Mix (Applied Biosystems, Foster City, CA, USA). Total reaction volume of 10 µL was reached with PCR-grade water. For tet(W), the primers used were tetW-F and tetW-R with tetW-S probe (Walsh et al., 2011). The reaction mix for tet(W) consisted of 2 µL of DNA template (between 0.08 and 1.39 ng/µL for seed bank DNA and 15 ng/µL for total community DNA), 0.025 µM of each primer, 0.1 µM of TaqMan probe and 1 × TaqMan®Fast Universal PCR Master Mix (Applied Biosystems, USA). Total reaction volume of 10 µL was reached with PCR-grade water. The qPCR program was the same for both genes and started with a hold at 95 °C for 10 min, followed by 45 cycles of denaturation at 95 °C for 15 s and annealing/elongation at 60 °C for 1 min. The qPCR assays were performed in technical triplicates on samples, standards and negative controls. The negative controls consisted of PCR blanks with only the reaction mix and of PCR blanks containing the mix and 2 µL of PCR-grade water. Standard curves were prepared from serial 10-fold dilutions of plasmid DNA containing the respective target gene in a range of 5 × 107 to 50 gene copies. For sul1, control plasmids and standard curves were prepared as previously described (Heuer & Smalla, 2007). For tet(W), standard curves were prepared as previously described (Walsh et al., 2011). The effect of inhibitors on amplification was tested for all the samples and for both genes. All samples were spiked with 104 copies of plasmid DNA containing the tet(W) or the sul1 gene and amplified together with the same set of non-spiked samples and control DNA and the results indicated that inhibition was negligible.

Sequencing and data analysis

Purified DNA extracts were sent to Fasteris (Geneva, Switzerland) for 16S rRNA amplicon sequencing using Illumina MiSeq platform (Illumina, San Diego, CA, USA), generating 250 bp paired-end reads. The hypervariable V3–V4 region was targeted using universal primers Bakt_341F (5′-CCTACGGGNGGCWGCAG-3′) and Bakt_805R (5′-GACTACHVGGGTATCTAATCC-3′) (Herlemann et al., 2011). Analysis of the dataset was made using Mothur (Schloss et al., 2009) following the standard MiSeq SOP (Kozich et al., 2013). The SILVA NR v123 reference database (Quast et al., 2013) was used for the alignment of amplicons and the taxonomic assignment of representative OTUs. After quality filtering and removal of chimeras, a total of 2’837’393 amplicons was obtained (625’339 unique sequences). Singletons were removed prior to the clustering into OTUs. The number of singletons in the dataset was 560’158. Clustering of the 2’277’235 remaining sequences (65’181 unique sequences) was made using a threshold of 97% identity. Finally, 11’802 OTUs constitute the dataset. The generated datasets were submitted to NCBI under the Bioproject accession number PRJNA396276.

Statistical and multivariate analyses

Community and statistical analyses were performed using R version 3.4.0 (R Core Team, 2014) and the phyloseq and vegan packages (McMurdie & Holmes, 2013; Oksanen et al., 2017). Pairwise correlations between OTU relative abundances and ARGs frequency were calculated using Spearman’s rank correlation coefficient. The same analysis was performed using the iron/manganese ratio as a proxy to lake eutrophication. Seed bank community was analyzed by principal coordinates analysis (PCoA), based on Bray–Curtis dissimilarity and Hellinger transformation of the OTUs table (community matrix). Environmental parameters and ARGs abundance/frequency were standardized and passively fitted to the ordination. Only significant parameters were displayed (p < 0.05).

Results

Quantification of ARGs in seed bank communities from sediment samples

Seed bank DNA was extracted from a sediment core previously validated for paleoecology covering approximately the last hundred years of sediment accumulation in Lake Geneva (Wunderlin et al., 2014a). ARG in seed bank DNA was measured by quantifying the number of copies of genes conferring resistance to tetracycline (tet(W) gene) and sulfonamide (sul1 gene), two commonly reported antibiotics detected in environmental settings (Davies & Davies, 2010). ARG quantification was standardized to DNA yield instead of number of 16S rRNA gene copies given the changes in community composition over time (see next section), and the variable number of copies of this molecular marker in different taxonomic groups (Lee, Bussema & Schmidt, 2009). The detection of ARGs in the seed bank DNA changed beginning in 1960 (tet(W)) and 1970 (sul1). However, the accumulation pattern was different for the two ARGs. In the case of tet(W), the total abundance of the gene (copies/g of sediment) increased by an order of magnitude since 1965 compared to the values obtained from 1920 to 1960 (Fig. S1). Moreover, the relative abundance of this ARG (gene copies/ng of DNA) in the seed bank DNA increased from 1961 to 1975 (Fig. 1). In the case of sul1, a steady increase of this ARG abundance was observed after 1970 (Fig. S1). The relative abundance of sul1 in seed bank DNA increased from the same period, followed by a decline and a more recent increase after the year ca. 2000 (Fig. 1). The specific timeframe in which enrichment in ARG counts per ng of DNA was observed concerned mainly the seed bank DNA, as opposed to the total bacterial community. In addition, we could detect ARGs using a lower initial concentration of DNA for the seed bank community (2 ng of DNA) compared to the total community (10–15 ng of DNA). This further suggests a preferential enrichment of ARGs in seed bank bacteria compared to the overall environmental background.

Figure 1 Tetracycline and Sulfonamide resistance in total bacterial community and in the seed bank over time.

Relative abundance (gene copies/ng of extracted DNA) of two genes conferring resistance to (A) the antibiotics tetracycline (tet(W)) and (B) sulfonamide (sul1) in sediment samples covering the period between 1920 and 2010 in Lake Geneva, Switzerland. Quantification was made in DNA extracted from the seed bank (SB DNA) and total microbial community (total DNA).

Figure 2 Seed bank community composition in sediments from Lake Geneva.

(A) Contribution (relative abundance) of individual genera from the six most abundant bacterial phyla present in the sediment samples. (B) Principal coordinates analysis (PCoA) of the seed bank bacterial community showing the effect of lake eutrophication (Axis 1; depth vector) and the accumulation of ARG (tet(W) and sul1 vectors).

Characterization of the seed bank communities

Previous studies in Lake Geneva have shown a dramatic effect of human activity on the nutritional status of the lake. The lake became eutrophic between 1954 and 1986, and this modified the proportion of some members of the bacterial community in sediments (Wunderlin et al., 2014a). Eutrophication is partly related to the same human activities that also shaped the antibiotic era (for example, increased agricultural and livestock output and population pressure). Since changes in microbial community composition as well as the spread of ARG within populations can influence the record of antibiotic resistance, it was important to analyze seed bank community composition alongside ARG quantification. Representatives of six major bacterial phyla (Proteobacteria, Firmicutes, Actinobacteria, Planctomycetes, Chlamydiae, and Chloroflexi) were the main components of the bacterial seed bank community in sediments (Fig. S2; Fig. 2A). The overall community analysis revealed similarities in the community composition in samples with higher relative abundance of either tet(W) or sul1 (Fig. 2B). For the former, a significant contribution of OTUs belonging to the Phylum Firmicutes was observed, while in the case of sul1 no particular bacterial group was correlated with increased accumulation.

Figure 3 Correlation of specific OTUs to the relative abundance of ARGs in sediments.

(A) Spearman correlation coefficients calculated for the relative abundance of each individual OTU and ARG frequency at different depths. The correlation coefficients were plotted as a continuum for the non-Firmicutes seed bank community (dashed line) or the OTUs belonging to Fimicutes only (solid line). (B) Relative abundance of the ten most positively correlated OTUs with the relative abundance of each individual ARG.

Table 1 Correlation analysis between individual OTUs and relative abundance of tet(W) and sul1.

Top 10 most positively and negatively correlated OTUs. For tet(W) gene, mostly OTUs belonging to Firmicutes have been correlated to tet(W) abundance. In contrast, for sul1, OTUs correlated to sul1 abundance belong to many phyla.

Gene	OTU	Phylum	Genus	Correlation coefficient	
tet(W)	Otu00093	Firmicutes	Anaerobacterium	0.7890	
	Otu01612	Firmicutes	Lachnoclostridium	0.7391	
	Otu00262	Firmicutes	Clostridiaceae 1 unclassified	0.7136	
	Otu00528	Firmicutes	Clostridium unclassified	0.6990	
	Otu01577	Firmicutes	Ruminococcus 1	0.6791	
	Otu00084	Firmicutes	Ruminococcacea unclassified	0.6722	
	Otu00908	Firmicutes	Ruminococcacea unclassified	0.6684	
	Otu02280	Firmicutes	Epulopiscium	0.6684	
	Out01131	Verrucomicrobia	Verrucomicrobiales unclassified	0.6659	
	Otu00529	Firmicutes	Geobacillus	0.6652	
sul1	Otu00318	Actinobacteria	Mycobacterium	0.6656	
	Otu00382	Chloroflexi	Caldilineaceae unclassified	0.6517	
	Otu00975	Firmicutes	Ruminiclostridium 1	0.6479	
	Otu03004	Firmicutes	Symbiobacterium	0.6341	
	Otu03302	Actinobacteria	Actinobacteria unclassified	0.6195	
	Otu00155	Proteobacteria	Hypomicrobium	0.6176	
	Otu00604	Verrucomicrobia	Verrucomicrobia unclassified	0.6170	
	Otu00853	Acidobacteria	Subgroup 6 unclassified	0.6103	
	Otu02777	Actinobacteria	Tessaracoccus	0.6095	
	Otu01652	Planctoymcetes	Plactomycetaceae unclassified	0.6092	

In order to understand more clearly the relationship between ARG enrichment and seed bank bacterial community, we next studied if the relative abundance of certain OTUs was correlated with ARG levels. For this, we calculated the correlation coefficient between the relative abundance of each OTU and the ARG relative abundance at different depths. Correlation coefficients were plotted as a continuum to analyze the overall response of the community (Fig. 3A). In the case of tet(W) most of the non-Firmicutes seed bank community was not correlated with increased ARG relative abundance over time (most correlation coefficients were close to 0; Fig. 3A; dashed line). However, when the analysis is made only for representatives of the Phylum Firmicutes, the distribution shifted significantly towards positive correlations (comparison of the distribution for the total and Firmicutes communities; t = 16.52, df = 6171.6, p-value < 2.2e−16; Fig. 3A; solid line). This analysis confirmed the results of the total community analysis (Fig. 2B). We investigated further the ten most positively correlated OTUs. Nine out of the ten operational taxonomic units (OTUs) positively correlated with tet(W) relative abundance belong to Firmicutes (Table 1). The origin and ecology of bacteria related to those OTUs suggests an equal contribution of bacteria from an environmental origin, mainly cellulose-degrading anaerobic bacteria such as Anaerobacterium (Horino, Fujita & Tonouchi, 2014) (OTU00093 and OTU00528), Clostridium (Hernandez-Eugenio et al., 2002; Miller et al., 2011; Zhilina et al., 2005) (OTU00262, OTU00084, and OTU02280), and Acetivibrio (Patel et al., 1980) (OTU00908); and from human (or animal) intestinal origin such as Ruminoccous (Cann, Bernardi & Mackie, 2016; Chassard et al., 2012; Crost et al., 2016) (OTU01612 and OTU01577). The OTUs positively correlated to tet(W) represented a minor fraction of the bacterial seed bank community even for those samples with the highest ARG abundance (relative OTU abundance not higher than 5%; Fig. 3B).

The same analysis performed on sul1 showed a larger fraction of the community positively correlated to relative ARG abundance (Fig. 3A), but in contrast to tet(W) this is not specifically significant for Firmicutes only. Instead, the 10 most positively correlated OTUs belonged to diverse phylogenetic groups (Actinobacteria, Chloroflexi, Firmicutes, Proteobacteria, Verrucomicrobia, and Planctomycetes) (Table 1). OTUs correlated positively with sul1 abundance represented only minor fractions of the seed bank community (Fig. 3A). Interestingly, the correlation coefficients are higher for tet(W) than for sul1, suggesting a stronger relationship of particular OTUs with the former.

Even though the analysis of the total community suggests that the effect of increased relative abundance of ARG appears to be independent from the generalized effect of eutrophication, we performed the same correlation analysis between relative OTU abundance and the iron/manganese ratio in sediments. The ratio of iron and manganese can be used as a proxy for redox conditions in the water column (Corella et al., 2012; Koinig et al., 2003) and changes in the relative concentration of these two elements have been shown to correlate with eutrophication in Lake Geneva (Wunderlin et al., 2014a). Eutrophication in Lake Geneva is one of the environmental disturbances with the best ecological record. Long-term trends show a steady increase of total phosphorus since 1957 with a peak in 1979. These values, together with phosphate data since 1970, indicate a shift in trophic status of the lake from oligotrophic to eutrophic taking place in the late 1960s. The system has since recovered, even though total phosphorus levels are still double the values before 1960 (Lazzarotto & Klein, 2012). The results show no overlap between the overall effect of eutrophication in specific OTUs (Fig. S3) and the effect of ARG abundance in terms of the most correlated OTUs (Fig. 3).

Discussion

Lake Geneva is one of the largest lakes in Europe and constitutes a major reservoir of drinking water. The composition of bacterial communities (Haller et al., 2011; Sauvain et al., 2014), as well as the presence of toxic metals (Pote et al., 2008), micropollutants (Bonvin et al., 2011), and ARGs (Czekalski et al., 2012; Czekalski, Gascon Diez & Bürgmann, 2014; Devarajan et al., 2015), has been monitored regularly in its water column and sediments. All these studies have demonstrated the role of human activity in the transfer of contaminants (including antibiotics) into sediments. Because of these preliminary studies, Lake Geneva is an ideal model system to validate the use of the seed bank bacterial community as a proxy to the effect of the historical use of antibiotics on the abundance of ARGs in the environment. Our results show that studying the bacterial seed bank community in sediments of Lake Geneva shows the historical increase in ARG abundance. There was a clear link between seed bank taxonomy and accumulation of tet(W). This taxonomy-specific effect has been well documented in the case of tetracycline (Roberts & Schwarz, 2016). Tetracycline is a class of broad-spectrum antibiotics active against a wide range of bacteria, including some atypical pathogens such as Mycoplasma and Chlamydia, and even eukaryotic parasites. In the USA, tetracycline became extensively used in production of livestock between 1950s and 1970s and remains today the second most commonly used antibiotic in agriculture (Roberts & Schwarz, 2016). The situation in Switzerland is similar, according to a recent report from the Swiss Federal Office of Public Health indicating that tetracycline (together with penicillin) is the second most sold antibiotic product, after sulfonamides (FOPH, 2016). In Switzerland, the current use of tetracycline is mainly restricted to non-medical applications, with a reported consumption below 1% in hospitals (according to data covering the period from 2004 to 2015) and close to 11% in outpatient settings (FOPH, 2016). In Switzerland the principal medical use of tetracycline was reported for the period of 1955–1970 (Table S1), but has since reduced dramatically following the use of amoxicillin-clavulanate for skin and soft-tissue infections and the increased use of cotromixazole (a combination of sulfonamides and trimethoprim) for uncomplicated urinary tract infections, which represent the two most common bacterial infections encountered in outpatient clinics and private medical practice.

Tetracycline binds to the elongating ribosome, affecting translation, and therefore resistance can be acquired through diverse mechanisms (Davies & Davies, 2010; Roberts & Schwarz, 2016). tet(W) is one of a series of ARGs conferring resistance through ribosomal protection and although the ancestral source of the gene is unknown, it has been reported in both Gram-positive and Gram-negative bacteria (Roberts & Schwarz, 2016). Our analysis suggest that medical historical use (1995–1970) fits well with the observed peak of relative accumulation of tet(W) in the seed bank DNA, which was highly correlated with changes in the abundance of Firmicutes. One potential explanation for the link between medical use of tetracycline and tet(W) in Fimicutes is the fact that the human gut microbiome can serve as a reservoir of ARGs, and in particular to genes conferring resistance to tetracycline (De Vries et al., 2011; Van Schaik, 2015). A recent analysis of the human gut microbiome suggests that Firmicutes are highly prevalent (Browne et al., 2016; Dethlefsen, McFall-Ngai & Relman, 2007). More importantly, a recent study suggests that sporulation is a widespread characteristic of the human microbiome (Browne et al., 2016), and it is precisely these dormant forms that can contribute to the seed bank in human-impacted ecosystems. However, linking tet(W) abundance and the human microbiome must not be seen as a confirmation of the relationship between medical antibiotic use and increase of ARGs levels in the environment. For example, a recent study monitoring the effect of tetracycline on the performance of anaerobic digesters used in wastewater treatment has also shown a highly significant increase in the relative abundance of spore-forming Firmicutes after treatment with a concentration of 20 mg/L of tetracycline (Xiong, Harb & Hong, 2017). Overall the data suggest that antibiotics such as tetracycline might select for specific groups of Firmicutes that can be later found in the seed bank archives.

The same analysis performed on sulfonamides, another class of antibiotics with an industrial history, shows a different trend. Sulfonamide drugs were also among the earliest antibiotics discovered. The legacy of mass production of sulfonamides is reflected in one of the most broadly disseminated cases of drug resistance, both in terms of prevalence and taxonomy (Aminov, 2010). Resistance to this class of antibiotic is almost universally associated to genetic mobile elements that confer a fitness advantage to the recipient bacteria as shown in the case of non-pathogenic Escherichia coli (Enne et al., 2004). The abundance of sul1 may thus be indicative of a dissemination trend of certain widespread mobile genetic elements (e.g., class-1 integrons) (Gillings, 2014; Skold, 1976; Skold, 2000) that may well carry other resistance elements. Horizontal gene transfer mediated by mobile genetic elements is considered a major pathway of ARG dissemination (Bengtsson-Palme, Kristiansson & Larsson, 2017; Berglund, 2015). This particular mechanism of ARG dissemination overcomes taxonomic barriers, probably explaining the wide taxonomic spectrum of bacterial seed bank groups correlated to sul1 quantification in the sediments.

The quantification of sul1 in the sedimentary record in the 1970s matches early prescription history of this antibiotic class (Table S1). More recent detection could be correlated to changes in guidelines to reduce usage of penicillin derivatives (such as co-amoxicillin) for uncomplicated urinary tract infection in favor of cotrimoxazole (Sulfamethoxazol-Trimethoprim combination), which may partially explain the common occurrence of sul1 resistance gene in the seed bank DNA especially after 2005 (Table S1). At this time medical guidelines changed given the high rate of resistance of E. coli (90% of the etiology of cystitis in healthy adult female humans) to penicillin derivatives, leading to the reintroduction of sulfonamides. Indeed, the resistance rate of E. coli to amoxicillin and to amoxicillin-clavulanate respectively reached 52% and 23% of the isolates tested at the Lausanne University Hospital Diagnostic Laboratory in 2016 (4,581 strains), which has prompted clinicians to use sulfonamides instead.

Conclusions

Previous studies of the historical legacy of the antibiotic era have come to contradictory conclusions. On the one hand, they show the recent effect of human activity on ARGs in the environment (Graham et al., 2016; Knapp et al., 2010; Thevenon et al., 2012), and suggest that reducing non-therapeutic antibiotic use may reduce some of the environmental ARG legacy. On the other hand, the results show that this is not universally applicable to all antibiotic classes and that policies intended to reduce non-therapeutic use can have undesirable consequences (Graham et al., 2016). Results for the accumulation of beta-lactamase genes in soils suggest that accumulation in soil reflected a broader expansion of antibiotic use across society, implying that development of resistance in clinical and agricultural systems is mutually influential (Graham et al., 2016). Our results generate valuable information for the debate regarding the long-term effect of the antibiotic era as we show that antibiotics also affect a fraction of the microbial community that will certainly outlast many of these policies: the seed bank bacterial community. This opens up a new debate, concerning the potential long-term effect of these dormant and persistent cellular structures and their potential for further spreading of ARGs in the environment. Importantly however, we hereby provide a proof of concept for a new way to study the historical development of resistance that is applicable to many geographic regions and resistance determinants and that does not rely on human archiving of environmental samples.

Supplemental Information

Figure S1 Total abundance of tetracycline and sulfonamide in sediment

Total abundance (gene copies/g of sediment) of two genes conferring resistance to the antibiotics tetracycline (tet(W)) and sulfonamide (sul1) in sediment samples covering the period between 1920 and 2010 in Lake Geneva, Switzerland. Quantification was made in DNA extracted from the seed bank (SB DNA) and total microbial community (total DNA).

Click here for additional data file.

Figure S2 Representation of the overall bacterial seed bank community composition evaluated to the taxonomic level of Phylum

Click here for additional data file.

Supplemental Information 1 Relative abundance of the ten OTUs most correlated with the iron/manganese ratio in the sediment samples used as proxies to lake eutrophication

Click here for additional data file.

Table S1 Temporal scale showing the respective period when a new antibiotic has been discovered, main period of clinical usage and the approximate year when a first resistance to that compound has been documented

The table is partially adapted from multiple sources (Clatworthy et al 2007, Torok et al 2009, van Hoek et al 2011), including national and international guidelines, as well as personal communication with Swiss and French doctors.

Click here for additional data file.

Supplemental Information 2 Raw data to generate Fig. 1 and Fig. S1

Click here for additional data file.

Additional Information and Declarations

Competing Interests

Author Contributions

DNA Deposition

Data Availability

The authors declare there are no competing interests.

Laura Madueño, Zhanna Bayrychenko and Karin Beck performed the experiments, contributed reagents/materials/analysis tools, reviewed drafts of the paper.

Christophe Paul conceived and designed the experiments, performed the experiments, analyzed the data, contributed reagents/materials/analysis tools, wrote the paper, prepared figures and/or tables, reviewed drafts of the paper.

Thomas Junier performed the experiments, analyzed the data, contributed reagents/materials/analysis tools, wrote the paper, prepared figures and/or tables, reviewed drafts of the paper.

Sevasti Filippidou performed the experiments, analyzed the data, contributed reagents/materials/analysis tools, wrote the paper, reviewed drafts of the paper.

Gilbert Greub conceived and designed the experiments, contributed reagents/materials/analysis tools, wrote the paper, reviewed drafts of the paper, historical data.

Helmut Bürgmann and Pilar Junier conceived and designed the experiments, contributed reagents/materials/analysis tools, wrote the paper, reviewed drafts of the paper.

The following information was supplied regarding the deposition of DNA sequences:

GenBank: PRJNA396276.

The following information was supplied regarding data availability:

The raw data has been provided as a Supplemental File.

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
