# Peer review of "A historical legacy of antibiotic utilization on bacterial seed banks in sediments"

_PeerJ, doi:10.7717/peerj.4197_

## Round 0.1 · original submission · Minor Revisions

Please provide a point-by-point response to all the reviewers comments.

Reviewer 1 ·

Basic reporting

• The paper is overall well written in a clear language. There are a few minor errors (see general comments below).
• The introduction relies heavily on a relatively small number of references (particularly Taylor et al 2011), and widening the perspective slightly would not hurt. For example, all the work of Fernando Baquero and José Martinez on environmental antibiotic resistance is entirely left without reference.
• The figures are relevant, but Table 2 seems inappropriate and contains much information that does not concern this manuscript. In Figure 2B, it is however unclear how the years were imposed on the PCoA.
• I could not find the raw data for the chemical analysis of iron and manganese associated with the manuscript.

Experimental design

• The study is original and within the scope of PeerJ.
• The manuscript aims to investigate if dormant microbial seedbanks in lake sediments can be used to track the abundance of antibiotic resistance genes in the environment over time. This is an important question for retrospective monitoring and such analysis would fill an important knowledge gap regarding baseline for antibiotic resistance prevalence in the environment.
• Although the scope is small (onty two genes – sul1 and tetW – are investigated) this is an important proof-of-concept study, and it conforms to methodological standards.
• However, the qPCR for the non-seedbank community would need a better description.
• Furthermore, it must be described how the chemical analysis for Mn and Fe in the sediments was carried out. Now, this analysis is just mentioned in passing on lines 241-242 in the manuscript.

Validity of the findings

• The data underlying the results seem to be robust and statistically sound.
• However, I have a major problem with one of the conclusions by the authors. The idea that tet(W) abundance would be a result of antibiotic pollution (or pollution by human bacteria carrying tet(W) genes) seem quite farfetched to me, given that the lake at the same time (according the authors) underwent eutrophication. Since eutrophication would lead to more anaerobic conditions, and the bacteria co-enriched with tet(W) are obligate anaerobes it seems that the explanation to the tet(W) increase would be increased abundances of these bacteria – not a selection for tet(W) by antibiotic residues. Indeed, some of these bacteria are known to harbour the tet(W) gene. If the authors want to make claims along the lines of “tetracycline exposure caused selection for tet(W) in the sediment communities”, they need to support this hypothesis with chemical measurements of tetracycline in the sediments.
• The paragraph on lines 278-301 is entirely speculative, but is presented as if it was well supported by the results. This needs to be modified.

Additional comments

• Line 19-20. It is unclear from the abstract why legacy of antimicrobial agent disposal is important.
• Line 41: “mitigates” seems to be me to be a weird choice of word in this context.
• Line 45-46: As far as I know the idea that ARGs are a fundamental component of microbial ecosystems is a hypothesis (albeit plausible), and should be presented as such rather than as a fact.
• Line 50-51: I don’t understand what the authors mean with “further hone naturally occurring antibiotic resistance systems”. Maybe it is the use of the word “hone” that confuses me.
• The entire first paragraph of the Introduction mixes the clinical/human society and environmental components of antibiotic resistance development in a quite confusing way, and particularly the lines 51-55 about risks really suffer from this problem. I would recommend the authors to read a recent review in FEMS microbiology (https://doi.org/10.1093/femsre/fux053), which may help in structuring the Introduction better.
• Line 56-58: I think that studies by Gerry Wrights lab make an even more convincing case for that ARGs were present before the antibiotic era, particularly these two papers: https://doi.org/10.1038/nature10388 & https://doi.org/10.1371/journal.pone.0034953
• Line 59: “in” -> “over”
• Line 71: “made” -> “make”
• Line 71: I think it is overstating the importance of sediments by attributing it “a highly likely source of resistance determinants” just because it is diverse. I would recommend that the language of this sentence is toned down.
• Line 79: “or” -> “and”
• Line 153: Which version of the SILVA database? This matters for reproducibility.
• Line 163: “OTUs relative abundance and ARGs frequency” -> “OTU relative abundances and ARG frequencies”
• Line 188: Where in the methods section is the total community DNA extraction protocol described?
• Line 198: “ARG” -> “ARGs”
• Line 239-240: I disagree with the authors here. To me, it looks like the observed changes in tet(W) could be entirely due to the effect of eutrophication (i.e. the increase of anaerobic bacteria). I would rather say that the results of the total community composition imply that eutrophication would be the cause of the tet(W) changes (although it is of course not the only possible explanation)
• Line 241-242: Where did the Fe and Mn data come from? How was those levels determined?
• Line 255: Insert “make it” after “Lake Geneva”.
• Line 257: “ARG” -> “ARGs”
• Line 278-280: This sentence is pure speculation, and it should be stressed as such. Actually, I would refrain from making this connection at all, given the vast number of possible confounding factors.
• Line 285-301: This entire paragraph is just pure speculation. If the authors want to make a claim that tetracycline has selected for tet(W), they should also quantify antibiotic residues in the sediment. The correlation between medical use and ARGs in sediments in a single lake is not a very convincing argument.
• Line 312-313: I disagree with that HGT would be a major pathway for dissemination of ARGs in aquatic environments. There are many environments that are more likely to be hotbeds for resistance transfer. However, waterways are great for dissemination of resistant bacteria, but that has nothing to do with HGT. I recommend the authors to read the review by Baquero et al (https://doi.org/10.1016/j.copbio.2008.05.006) or the aforementioned FEMS Microbiology review (https://doi.org/10.1093/femsre/fux053) to straighten out these concepts, as they again seem to be intermixed here by the authors.
• Line 316-319: The relation to penicillin and cotrimoxazole here is an interesting speculation, but how come that sul1 has not been enriched earlier when sulfonamides where widely used in healthcare? This would need an explanation as well.
• Line 321-323: How is this last sentence related to sulfonamide resistance?
• Line 329: “can” -> “may”
• Line 336-337: I would rather spell it out as that the taxonomy of the seed bank community influences the ARG content.
• Line 338: Contaminated in what sense? Are these genes really contaminations if they allow better resilience to the bacterial community?
• Table 2 contains much information that does not concern this manuscript and could probably be left out.
• Figure 2B: How were the years imposed on the PCoA?

·

Basic reporting

The manuscript is nicely written and clear. It was very enjoyable to read through the study.

Experimental design

The research presented in the manuscript is original and interesting. Also, consistent with the Aims and Scope of PeerJ journal.

The research results are interesting and have a good contribution to study of antibiotic resistance in aquatic environment throughout the history of anthropogenic used of antibiotics.

Although the methods were described thoroughly, there are some information that would be better to be added in the material and methods, as follow:
1. Correlation of the depth of sediment core with the years of study (Line 98-102), also in Figure 1 and Supplementary Figure 1 (This information was only described in supplementary metadata)
2. Efficiency of DNA extraction from each sample (L 122)
3. Limit of quantification from the qPCR measurement of each gene (L 139)
4. The explanation to use the amount of DNA concentration as normalisation instead of using the amount of 16S rRNA gene

Validity of the findings

The finding is well reported and giving new contribution on providing insightful information to prevent the spread of ARGs in the environment. If there is one question, it is in the decision to study tetW gene as the representative of more than 20 types of tetracycline resistance genes that are known. Several studies shown that tetM gene is the most abundant of tetracycline resistance gene found in sediment environments.

---

## Round 0.2 · accepted · Accept

We thank you for the detailed and thorough revision and we hope you will consider PeerJ again for your future papers.